# PTP61F Mediates Cell Competition and Mitigates Tumorigenesis

**DOI:** 10.3390/ijms222312732

**Published:** 2021-11-25

**Authors:** John E. La Marca, Lee F. Willoughby, Kirsten Allan, Marta Portela, Pei Kee Goh, Tony Tiganis, Helena E. Richardson

**Affiliations:** 1Cell Polarity, Cell Signaling & Cancer Laboratory, Department of Biochemistry & Genetics, La Trobe Institute for Molecular Science, La Trobe University, Melbourne, VIC 3086, Australia; E.LaMarca@latrobe.edu.au (J.E.L.M.); Kirsten.OCallaghan82@gmail.com (K.A.); M.PortelaEsteban@latrobe.edu.au (M.P.); 2Cell Cycle & Development Laboratory, Research Division, Peter MacCallum Cancer Centre, Melbourne, VIC 3002, Australia; LeeFWilloughby@gmail.com; 3Monash Biomedicine Discovery Institute, Monash University, Clayton, VIC 3800, Australia; pei.goh@monash.edu (P.K.G.); tony.tiganis@monash.edu (T.T.); 4Department of Biochemistry and Molecular Biology, Monash University, Clayton, VIC 3800, Australia; 5Peter MacCallum Department of Oncology, Department of Anatomy & Neuroscience, Department of Biochemistry, University of Melbourne, Melbourne, VIC 3010, Australia

**Keywords:** PTP61F, RAS, JAK–STAT, cell competition, tumorigenesis

## Abstract

Tissue homeostasis via the elimination of aberrant cells is fundamental for organism survival. Cell competition is a key homeostatic mechanism, contributing to the recognition and elimination of aberrant cells, preventing their malignant progression and the development of tumors. Here, using *Drosophila* as a model organism, we have defined a role for protein tyrosine phosphatase 61F (PTP61F) (orthologue of mammalian PTP1B and TCPTP) in the initiation and progression of epithelial cancers. We demonstrate that a *Ptp61F* null mutation confers cells with a competitive advantage relative to neighbouring *wild-type* cells, while elevating PTP61F levels has the opposite effect. Furthermore, we show that knockdown of *Ptp61F* affects the survival of clones with impaired cell polarity, and that this occurs through regulation of the JAK–STAT signalling pathway. Importantly, PTP61F plays a robust non-cell-autonomous role in influencing the elimination of adjacent polarity-impaired mutant cells. Moreover, in a neoplastic RAS-driven polarity-impaired tumor model, we show that PTP61F levels determine the aggressiveness of tumors, with *Ptp61F* knockdown or overexpression, respectively, increasing or reducing tumor size. These effects correlate with the regulation of the RAS–MAPK and JAK–STAT signalling by PTP61F. Thus, PTP61F acts as a tumor suppressor that can function in an autonomous and non-cell-autonomous manner to ensure cellular fitness and attenuate tumorigenesis.

## 1. Introduction

Over the past century, *Drosophila melanogaster* has proven a suitable organism for modelling a range of human disorders, including cancer. The majority of the “hallmarks of cancer” are capable of being modelled in *Drosophila*, which, together with the high level of conservation of disease-relevant genes and its short life cycle, makes *Drosophila* a useful model for studying tumorigenesis (reviewed in [1,2,3]). One of the best-researched methods of tumorigenesis initiation in *Drosophila* is via the phenomenon of cooperative tumorigenesis—for example, upon the simultaneous activation of the growth-promoting GTPase RAS85D (commonly referred to as RAS, human orthologues HRAS, KRAS, and NRAS) and loss of the apico-basal cell polarity regulators Scribble (SCRIB) or Discs large 1 (DLG1) (human orthologues SCRIB and DLG1-4, respectively) (reviewed in [4]). Together, these alterations promote the formation of neoplastic, invasive tumors in developing *Drosophila* via the activation of anti-apoptotic signals, and contribute to the co-option of c-Jun N-terminal Kinase (JNK) signalling into a proliferation-promoting signalling pathway (reviewed in [5]). However, RAS85D activation or polarity impairment individually cause only “pre-tumorigenic” tissue disruptions—RAS85D activation leads to benign tissue overgrowth [6,7], while polarity impairment leads to increased cell proliferation and cell death, differentiation defects, and increased cell migration/invasion [7,8,9,10,11,12,13,14]. Interestingly, whole tissues depleted of SCRIB or DLG1 massively overgrow, but when polarity-impaired cells are generated in a clonal manner in *Drosophila* developing epithelial tissues (the wing or eye-antennal imaginal discs), *scrib* or *dlg1* mutant cells are subject to a tissue surveillance and homeostasis mechanism known as cell competition [10,15]. The core concept of cell competition is that cells within a tissue are intrinsically competing with one another—that is, the fitness of each cell is surveyed relative to their neighbouring cells, and cells that are less fit are actively eliminated to maintain tissue homeostasis (reviewed in [16]). This is a conserved mechanism, though it is not so well studied in mammals as it is in flies (reviewed in [17]). Colloquially, eliminated cells are referred to as “loser cells”, while those that eliminate and replace them are concomitantly termed “winner cells”. In *Drosophila* epithelial tissues, *scrib*/*dlg1* mutant clones undergo cell competition, and are eliminated by their *wild-type* neighbours [10,15]. Modulations in several highly conserved signalling pathways are involved in eliminating polarity-impaired cells during cell competition, including the JNK, the Hippo tissue growth inhibitory, the Janus kinase-signal transducer and activator of transcription (JAK–STAT), and the Epidermal Growth Factor Receptor (EGFR)-RAS–mitogen activated protein kinase (MAPK) signalling pathways (reviewed in [16]). The roles and targets of each cell competition-induced signalling pathway and the interplay between them are not completely understood, but many are regulated via the actions of protein tyrosine kinase (PTK) pathways, making the investigation of protein tyrosine phosphatases (PTPs) that regulate these pathways in cell competition a logical step.

Phosphorylation of proteins by protein kinases and dephosphorylation by protein phosphatases are critical for regulating the spatial and temporal activity of many signalling pathways. Protein phosphatases are broadly classified into three groups: protein serine/threonine phosphatases, PTPs, and dual-specificity phosphatases (DUSPs) (reviewed in [18,19,20]). While the majority of protein phosphorylation in mammalian cells occurs on serine/threonine, rather than tyrosine, tyrosine phosphorylation-dependent signalling is critical for the ability of cells to communicate with their extracellular environment and each other. More than 100 structurally and functionally diverse PTPs have been identified in the human genome [20]. In the vinegar fly model organism, *Drosophila melanogaster*, there are currently 44 known PTPs (including DUSPs), all with conserved human orthologues, though not all human PTPs have fly orthologues [21,22,23]. As in mammals, PTPs in *Drosophila* belong to either the transmembrane receptor or non-receptor subtypes [21]. The *Drosophila* non-receptor protein tyrosine phosphatase 61F (*Ptp61F*) functions as a negative regulator in a number of highly conserved signalling pathways, including the JAK–STAT pathway [24,25,26,27], the Insulin-like Receptor (INR) pathway [24,27,28,29], the EGFR- pathway [27,28], and the platelet-derived growth factor (PDGF)- and vascular endothelial growth factor (VEGF)-receptor-related (PVR) pathway [27]. PTP61F has two mammalian orthologues, PTP1B (encoded by *PTPN1*) and TCPTP (encoded by *PTPN2*), which share 74% catalytic domain sequence identity and 86% similarity (reviewed in [30,31]). PTP1B was the first mammalian PTP identified [32], and is localised to the cytoplasmic face of the endoplasmic reticulum and has an important role in immunity and metabolism, acting to dephosphorylate substrates such as the INR [33,34,35] and JAK-family PTKs JAK-2 and Tyk2 [36]. PTP1B can serve as a tumor suppressor, but also as an oncoprotein, as elevated PTP1B levels can contribute to the activation of SRC-family PTKs [37,38] and mediate signalling by the HER-2 oncoprotein [39,40]. *PTPN2* encodes two splice variants: A 48 kDa TCPTP (TC48) which, like PTP1B, is localised to the endoplasmic reticulum, and a 45 kDa variant (TC45) that is targeted to the nucleus but shuttles between the nuclear and cytoplasmic environments [41,42,43,44]. Consequently, TCPTP has access to both nuclear substrates, such as STAT-1/3/5, and cytoplasmic substrates, such as INR and JAK-1/3 [31,44]. Interestingly, TCPTP and PTP1B have both overlapping and distinct functions in mammals [31]. More generally, TCPTP is thought to serve as a tumor suppressor, particularly in T cell acute lymphoblastic leukemia [45,46], and possibly in breast cancer [47] and liver cancer [48,49]. However, the role of *Ptp61F* in *Drosophila* cancer models has not been explored.

This study therefore aimed to explore the role of *Ptp61F* in cell competition and cooperative tumorigenesis, examining how its up-/downregulation affects clonal growth and cell polarity-impaired RAS-driven tumor growth. We report that PTP61F acts as a negative regulator of the JAK–STAT signalling pathway and thereby coordinates cell survival and clonal growth during cell competition. Furthermore, we show that it acts as a negative regulator of RAS85D-driven polarity-impaired cooperative tumorigenesis and inhibits the RAS–MAPK and JAK–STAT signalling pathways that are normally hyperactive in this context.

## 2. Results

### 2.1. Ptp61F Impairment Confers a Competitive Advantage on Epithelial Clones

Various highly conserved well-characterised signalling pathways are involved in cell competition in *Drosophila* epithelial tissues, such as EGFR-RAS–MAPK and JAK–STAT signalling (reviewed in [16]). These pathways are variably orchestrated by reversible tyrosine phosphorylation. Therefore, we reasoned that PTPs might play an unappreciated role in cell competition. We selected the *Drosophila* tyrosine phosphatase *protein tyrosine phosphatase 61F* (*Ptp61F*) as a candidate, as it has been shown to negatively regulate EGFR-RAS–MAPK and JAK–STAT signalling [27].

First, we utilised a technique known as “twin-clone generation”, whereby a single recombination event simultaneously generates GFP-double-positive *wild-type* clones and GFP-negative mutant clones, in a background of GFP-single-positive *wild-type Drosophila* third-instar larval (L3) epithelial tissues (technique adapted from Froldi et al. [50]). In control L3 wing imaginal discs, generating twin-clones where both are *wild-type*, though one twin is GFP-double positive and the other is GFP negative (Figure 1A,A’), leads to clones where the GFP-double-positive clones are consistently slightly larger than their GFP-negative twins, perhaps due to background genetic effects (Figure 1E). However, when the GFP-negative clones are homozygous mutant for *Ptp61F* (Figure 1B,B’), using the null allele *Ptp61F^Δ^* [24], the trend is reversed and *Ptp61F^Δ/Δ^* clones are consistently larger than their *wild-type* twins (Figure 1E). Analysing the GFP-negative/GFP-double-positive size ratio for each twin-clone pair further revealed a statistically significant, ~30% increase in the size ratios between *wild-type/wild-type* and *wild-type/Ptp61F^Δ/Δ^* twin-clones in wing imaginal discs (N.B. for all statistical test results, see the corresponding figure legend).

These results are supported by similar findings in experiments using L3 eye-antennal imaginal discs, where *wild-type/wild-type* twins once again were observed to have a slightly larger GFP-double-positive twin (Figure 1C,C’,F), but *wild-type/Ptp61F^Δ/Δ^* twins have a consistently larger GFP-negative, *Ptp61F* mutant twin (Figure 1D,D’,F). Similar to the wing discs, the GFP-negative/GFP-double-positive size ratio was significantly larger in *wild-type/Ptp61F^Δ/Δ^* twins compared to *wild-type/wild-type* twins, by ~2-fold. Altogether, these data show that loss of *Ptp61F* confers a competitive advantage upon epithelial tissue clones, possibly by promoting cell survival and/or proliferation.

### 2.2. Ptp61F Regulates Polarity-Impaired Clone Survival/Growth during Cell Competition

One mode of cell competition in *Drosophila* occurs during the removal of polarity-impaired cells from larval epithelial tissues—*scrib* or *dlg1* mutant cells are actively outcompeted and eliminated from epithelial tissue by several different cell competition mechanisms and signalling pathways (reviewed in [16]). Using the genetic tools available in *Drosophila*, patches of mutant cells can be generated clonally in tissues of interest to model cell competition. Our twin-clone analyses suggested that PTP61F has a role in suppressing the ability of cells to compete, reducing their relative fitness and facilitating their elimination, as knocking-out *Ptp61F* clonally allows cells to outcompete their neighbours (Figure 1). We next asked whether *Ptp61F* reduction in cells that are relatively less fit (e.g., *scrib* mutant cells) might abrogate their elimination phenotype (Figure 2A). To investigate this, we utilised the mosaic analysis with a repressible cell marker (MARCM) technique, allowing for transgenes of interest and cell markers to be expressed in cells mutated for a gene-of-interest [51]. Using L3 eye-antennal imaginal discs, we examined how *Ptp61F* knockdown affected the growth of clones with mutant *scrib* (*scrib^1^*) that have a loser cell fate. In control discs, where both GFP-marked clones and the remainder of the tissue were otherwise *wild-type* (except that a *UAS-myr RFP* transgene was present in RNAi-free control samples as a *UAS* balancing element, and *UAS-Dcr-2* (a.k.a. Dicer) was also present in all samples), the induced clones make up ~40% of the tissue volume (Figure 2B,F). Expression of RNAi against *Ptp61F* (v37436) alone did not significantly alter this clonal tissue volume, with it remaining ~40% (Figure 2C,F) (RNAi efficacy demonstrated in Appendix A; *Ptp61F* mRNA levels reduced by ~60%). As expected, inducing clones homozygous for mutant *scrib* led to their contributing to a markedly smaller proportion of the total tissue volume: Around ~11% (Figure 2D,F). However, expression of RNAi against *Ptp61F* within those *scrib^1/1^* clones led to a small but statistically significant increase in their total volume: to an average of ~13% of the total tissue (Figure 2E,F). These data suggest that *Ptp61F* has a role, albeit small, in suppressing the ability of polarity-impaired cells to “fight back” against the efforts of neighbouring *wild-type* cells to eliminate them.

### 2.3. JAK–STAT Signalling Plays a Role in the Fitness of Scrib-Mutant Clones and Is Required Downstream of Ptp61F Knockdown for the Increased Survival of Scrib-Mutant Clones

Our data thus far have demonstrated a new role for PTP61F as contributing to polarity-impaired cell competition. Previous studies have shown that PTP61F can attenuate JAK–STAT signalling in *Drosophila* [24,25,26,27,52], but whether PTP61F regulates JAK–STAT signalling in the context of cell competition is unclear. Moreover, although JAK–STAT signalling is known to play a role in the *wild-type* winner cells during polarity-impaired cell competition [53], it is unclear whether JAK–STAT signalling has a role within the polarity-impaired loser cells. Therefore, using MARCM techniques, we investigated the requirement of JAK–STAT signalling in the competitiveness of polarity-impaired cells, and whether this occurs downstream of PTP61F.

We used RNAi against *Stat92E* (v43866) (RNAi efficacy demonstrated in Appendix A; *Stat92E* mRNA levels reduced by ~40%) to determine whether JAK–STAT signalling is necessary for *scrib*-mutant clone elimination. Eye-antennal discs expressing *Stat92E^RNAi^* had clones contributing to ~20% of the tissue volume, a significantly smaller fraction than the *wild-type* controls at ~40% (compare Figure 3A,C, quantified in Figure 3I), consistent with *Stat92E* influencing clonal fitness. Similarly, discs with *Stat92E^RNAi^*-expressing *scrib^1/1^* clones contributed to ~5% of the tissue volume, a significantly smaller proportion than the *scrib^1/1^* controls at ~11% (compare Figure 3E,G, quantified in Figure 3I). These data suggest that STAT92E functions within *scrib*-mutant clones during cell competition to oppose their elimination.

Next, we examined whether elevated JAK–STAT signalling was driving *scrib*-mutant clone growth suppression upon *Ptp61F* knockdown. To do so, we combined knockdown of *Stat92E* and *Ptp61F* in MARCM-generated clonal tissue. In otherwise *wild-type* clones, expression of both *Stat92E^RNAi^* and *Ptp61F^RNAi^* resulted in clones that contributed to ~20% of the tissue, a non-significant effect relative to the *Stat92E* knockdown alone (also ~20%; compare Figure 3C,D, quantified in Figure 3I). However, these clones were significantly smaller than the *Ptp61F^RNAi^*-only clones (tissue volume of ~40%; compare Figure 3B,D, quantified in Figure 3I). By contrast, in *scrib*-mutant clones (which contribute to 11% of the tissue), simultaneous *Ptp61F* and *Stat92E* knockdown led to clones contributing to ~7% of the tissue, a statistically significant decrease in clonal tissue volume compared to *scrib*-mutant *Ptp61F* knockdown-only clones at ~13% (compare Figure 3F,H, quantified in Figure 3I), revealing a requirement for *Stat92E* in the *Ptp61F*-knockdown-mediated rescue of *scrib*-mutant clone size. In comparison with *scrib*-mutant *Stat92E* knockdown-only clones, which make up only ~5% of the tissue, simultaneous *Ptp61F* and *Stat92E* knockdown in *scrib*-mutant clones resulted in a statistically significant increase to 7% in clonal tissue volume (compare Figure 3G,H, quantified in Figure 3I), showing that the presence of *Ptp61F* contributes to the competitive disadvantage of *scrib*-mutant *Stat92E* knockdown clones. Together, these data show that STAT92E levels influence the survival of *scrib*-mutant clones, and that the increased survival of *scrib*-mutant clones upon *Ptp61F* knockdown is dependent on *Stat92E*.

### 2.4. Ptp61F Regulates Wild-Type Clone-Mediated Elimination of Polarity-Impaired Clones during Cell Competition

Our results indicate that PTP61F is necessary within polarity-impaired cells to help promote their elimination. However, since JAK–STAT signalling in the *wild-type* “winner” cells neighbouring *scrib* mutant cells has been shown to be important for *scrib* mutant cell elimination [53], it was possible that PTP61F activity might also be involved in the competitive function of “winner” cells. To investigate the role of PTP61F in cells neighbouring polarity-impaired cells during competition, we utilised the reverse mosaic analysis with a repressible cell marker (revMARCM), which allows the generation of GFP-negative clones mutant for a gene-of-interest within a tissue expressing GFP together with other transgenes [51]. Using L3 eye-antennal imaginal discs, we used revMARCM to examine the effect of overexpression of a *Ptp61F* isoform on the growth of neighbouring clones that were homozygous mutant for the *scrib* allele, *scrib^1^* (Figure 4A). Control discs, where GFP-negative clones and the rest of the tissue was functionally *wild-type*, contained GFP-negative clones with an average volume of ~60% of the tissue (Figure 4B,F). By contrast, when GFP-negative clones formed in a background of tissue that was overexpressing the nuclear-localised form of *Ptp61F* (hereafter referred to as *Ptp61Fn*, expressed using the 2nd chromosome line *Ptp61Fn^2.1^*, the B isoform of PTP61F proteins, which is more potent than the A isoform) [24], the GFP-negative clones contributed to ~70% of the tissue, a statistically significant increase (Figure 4C,F). This suggested that PTP61Fn activity within clones reduces their fitness relative to their *wild-type* clone neighbours.

We next examined the effect of *Ptp61Fn* expression (GFP positive) on the clonal growth of neighbouring homozygous *scrib* mutant (*scrib^1^*) clones (GFP negative). As expected, when surrounded by *wild-type* tissue the GFP-negative *scrib*-mutant clones were reduced in volume and made up, on average, ~30% of the total volume (Figure 4D,F). *Ptp61Fn* expression (GFP positive) in the cells neighbouring the *scrib*-mutant clones led to a statistically significant increase in the *scrib*-mutant clone volume (GFP negative), to an average of ~50% of the total (Figure 4E,F). This further supports our assertion that PTP61Fn negatively impacts the ability of a cell to compete with their neighbours and, by extension, negatively affects the ability of otherwise *wild-type* cells to outcompete and eliminate *scrib*-mutant cells.

Using revMARCM techniques, we then also examined the effect of *Ptp61F* knockdown in the *wild-type* cells (GFP positive) surrounding *scrib*-mutant cells (GFP negative) (Figure 4A). In this instance, *wild-type* control tissues had a GFP-negative to total tissue volume ratio of ~70% (Figure 4G,K), while expression of *Ptp61F^RNAi^* in the GFP+ tissue led to a small, non-significant reduction in GFP-negative clone (*wild-type*) volume to ~60% (Figure 4H,K). Meanwhile, knockout of *scrib* (GFP negative) resulted here in GFP-negative clones contributing to ~45% of the total tissue (Figure 4I,K), but the introduction of *Ptp61F^RNAi^* expression in neighbouring cells (GFP positive) led to a significant reduction in that ratio to ~30% (Figure 4J,K). These data are consistent with our revMARCM results with PTP61Fn overexpression, and together show that PTP61F levels within winner cells strongly influence cell competition of polarity-impaired cells, more so than the modest effect when knocked down within the *scrib* mutant cells (15% compared with 2%). Most likely, PTP61F’s non-cell-autonomous effect is achieved by regulating JAK–STAT signalling activity in the *wild-type* cells, which is known to be necessary in winner cells for the elimination of adjacent *scrib*-mutant cells, and functions in a manner independent of cell proliferation [53].

### 2.5. Ptp61F Suppresses Activated Ras85D-Driven Polarity-Impaired Epithelial Tumorigenesis

To investigate whether *Ptp61F* in also involved in tumor progression, we utilised an activated RAS85D-driven polarity-impaired model of tumorigenesis [54]. In this model, controlled expression of a constitutively active form of RAS85D, known as RAS85D^V12^ (a.k.a. RAS^V12^ or RAS^ACT^), was combined with RNAi-mediated knockdown of the cell polarity regulator *dlg1* (a component of the Scribble cell polarity module) in the entire eye-antennal epithelium using *eyFLP; Actin >> GAL4* (*EAG*, and *EAGRD* when including *Ras85D^V12^* and *dlg1^RNAi^*). This leads to neoplastic tumor formation in the eye-antennal epithelium and organismal death as oversized (“giant”) larvae or pupae. Using this model, we examined the effect of RNAi-mediated knockdown of *Ptp61F* or overexpression of nuclear-localising *Ptp61Fn* on tumor growth (Figure 5A–D’) and *Drosophila* development (Figure 5E–H). Dissection of L3 eye-antennal imaginal disc-brain complexes revealed that, as previously described [54], the eye-antennal tissue of *EAGRD > GFP* larvae exhibited neoplastic overgrowth compared to the *EAG > GFP* control (Figure 5A,B) and led to developmental delay/arrest, primarily at the early pupal stage (Figure 5E,G). Expression of RNAi targeting *Ptp61F* in *EAGRD>GFP* larvae resulted in increased tumor size (Figure 5C,C’) and a developmental delay/arrest, with the majority of animals observed being arrested at a “giant” larval stage (Figure 5E,F). In contrast, *Ptp61Fn* expression substantially suppressed tumor growth (Figure 5D,D’) and alleviated the developmental delay/arrest, as the majority of individuals reached the late pupal stage (Figure 5E,H). Notably, control animals expressing *Ptp61F^RNAi^* or *Ptp61Fn* via the *EAG* system did not appear to suffer any developmental issues and produced adults at expected frequencies (Figure 5E). Therefore, these results suggest that PTP61F suppresses activated RAS85D-driven polarity-impaired epithelial tumorigenesis in *Drosophila*.

### 2.6. Drosophila Ptp61F Represses pTyr, RAS–MAPK, and JAK–STAT Signalling in Activated Ras85D-Driven Polarity-Impaired Tumors

To investigate the signalling pathways regulated by PTP61F in its suppression of activated *Ras85D*-driven polarity-impaired epithelial tumorigenesis, we first examined the effects of *Ptp61F* overexpression or knockdown on the activation of tyrosine phosphorylation-dependent signalling pathways. To this end we immunoblotted *EAGRD* tumor lysates (using dissected L3 eye-antennal imaginal discs) with an antibody that detects tyrosine phosphorylated proteins (pTyr). Consistent with the established role of PTP61F as a tyrosine phosphatase, the expression of *Ptp61Fn* resulted in markedly decreased tyrosine phosphorylation, relative to the control, whereas *Ptp61F* knockdown (via *Ptp61F^RNAi^*) increased the overall tyrosine phosphorylation observed (Figure 6A,B).

We then assessed whether RAS–mitogen-activated protein kinase (MAPK) signalling, a PTK-activated pathway (reviewed in [55,56]), might also be affected by PTP61F overexpression or knockdown in RAS-driven polarity-impaired tumors. Using an antibody specific for the phosphorylated and activated form of the mammalian MAPK ERK1/2 (pERK), we observed that eye-antennal tissue from *EAGRD* larvae had substantially upregulated pERK (relative to total ERK) when compared to the *EAG* control (Figure 6C,D, compare lanes 3 and 6). Consistent with our previous findings [27], we found that the overexpression of PTP61Fn decreased pERK (by ~75%, Figure 6C,D, compare lanes 1 and 3), while *Ptp61F* knockdown increased pERK (by ~25%, Figure 6C,D, compare lanes 2 and 3). These data suggest that PTP61F attenuates RAS–MAPK signalling in *Ras85D**^V12^*-driven polarity-impaired tumors.

Since PTP61F has been shown to negatively regulate JAK–STAT signalling in *Drosophila* tissues [24,25,26], and polarity-impaired tissues upregulate JAK–STAT signalling [15,57,58], we then asked whether STAT92E expression and signalling was repressed by PTP61F in activated *Ras85D**^V12^*-driven polarity-impaired eye-antennal epithelial tissue. To monitor JAK–STAT signalling we used the *Stat92E*-binding site *lacZ* reporter (*Stat92E-lacZ*) [59] and analysed *EAGRD* tumorous eye-antennal epithelial tissue for β-galactosidase (β-gal) expression via Western blotting (Figure 6C,D). Using this system, we found that in *EAGRD* and *EAG* eye-antennal tissue, the expression of *Ptp61Fn^2.1^* substantially repressed β-gal levels (to ~50%, Figure 6C,D, compare lanes 1 and 3). However, we found that the *Ptp61F^RNAi^*-mediated knockdown of PTP61F was not sufficient to increase β-gal expression (Figure 6C,D, compare lanes 2 and 3). Taken together with our data on the effect of PTP61F on pERK levels in these tumorous tissues, these data show that PTP61F inhibits both JAK–STAT and RAS–MAPK signalling to attenuate *EAGRD*-driven tumorigenesis.

To further assess the effects of PTP61F on JAK–STAT signalling in situ, we used the *equatorial-GAL4* (*eq-GAL4*) system to express *Ras85D^V12^* and *dlg1^RNAi^* in a patch of cells in the centre of the developing eye epithelium (marked with *UAS-RFP*), coupled with an endogenous reporter of JAK–STAT signalling, *10×Stat92E-GFP* [60]. As expected, *Ras85D^V12^* and *dlg1^RNAi^* expression robustly increased (by ~3-fold) *Stat92E-GFP* expression in the *eq* domain, as well as expanded the relative domain size indicating that tissue growth was stimulated (compare Figure 7B,B’,B’’ and Figure 7A,A’,A’’, quantified in Figure 7E,F). Strikingly, expression of *Ptp61Fn^4.1^* (3^rd^ chromosome line) in *Ras85D^V12^*- and *dlg1^RNAi^*-expressing tissue significantly decreased (by ~25%) *Stat92E-GFP* expression in, and reduced the relative area of, the *eq* domain indicating that tissue growth was inhibited (compare Figure 7C,C’,C’’ and Figure 7B,B’,B’’, quantified in Figure 7E,F). To reduce the level of PTP61F, rather than with RNAi, we used the *Ptp61F^Δ^* null allele [24]. Flies heterozygous for *Ptp61F^Δ^* (therefore expecting a reduction in PTP61F levels of 50% in the mutant tissue as well as *wild-type* tissue) and expressing both *Ras85D^V12^* and *dlg1^RNAi^* in the *eq* domain resulted in a non-significant increase in *Stat92E*-GFP levels, although the relative *eq* domain size was significantly increased (compare Figure 7D,D’,D’’ and Figure 7B,B’,B’’, quantified in Figure 7E,F), indicating that tissue growth of the tumorous tissue was increased relative to the *wild-type* tissue. Consistent with the Western blot analyses (Figure 6C,D), these data demonstrate that increased *Ptp61F* expression represses JAK–STAT signalling and tissue growth in activated *Ras85D*-driven polarity-impaired epithelial tumors.

## 3. Discussion

### 3.1. Summary of the Result of This Study

In this study, we have examined a role for PTP61F, the *Drosophila* orthologue of the mammalian PTPs PTP1B and TCPTP, in pre-tumorigenic and tumorigenic tissue models. We observed that the loss of *Ptp61F* allows otherwise *wild-type* clones to outcompete their *wild-type* neighbours, and *Ptp61F* knockdown allows polarity-impaired “loser” cell clones to overcome some of their relative fitness disadvantage. We then demonstrated that the competitive advantage conferred on polarity-impaired mutant clones by *Ptp61F* knockdown is dependent on *Stat92E*. Furthermore, PTP61F can also function non-cell autonomously; overexpression of *Ptp61F* in “winner” cells increases the survival of the neighbouring polarity-impaired clones, whereas *Ptp61F* knockdown in the *wild-type* “winner” cells decreases the survival of polarity-impaired cells. This non-cell-autonomous function of PTP61F has a considerably stronger effect upon polarity-impaired clonal survival than its cell-autonomous function within the polarity-impaired cells. We also show that in larvae possessing *Ras85D^V12^*/*dlg1^RNAi^*-driven tumors, *Ptp61F* knockdown promoted tumor growth and suppressed organism development, while PTP61F overexpression suppressed tumor growth and partially rescued the defective development of the tumorous animals. We demonstrated that, in *Ras85D^V12^*/*dlg1^RNAi^*-driven tumorigenic tissues, *Ptp61F* knockdown increased levels of tyrosine phosphorylated proteins and RAS–MAPK signalling, while *Ptp61F* overexpression attenuated tyrosine phosphorylation, and RAS–MAPK and JAK–STAT signalling. Altogether our results suggest that *Ptp61F* acts as a tumor suppressor, limiting the competitive ability of cells, and downregulating the cell survival and proliferation otherwise driven by the RAS–MAPK and JAK–STAT signalling pathways.

### 3.2. PTP61F in Signalling Pathway Regulation during Cell Competition

We have shown, along with previous studies [24,25,26,27,28], that PTP61F is capable of downregulating the JAK–STAT and RAS–MAPK signalling pathways. Although well established as evolutionarily conserved regulators of cell proliferation and tissue growth, JAK–STAT and RAS–MAPK signalling are also involved in polarity-impairment-induced cell competition (reviewed in [16]). During both cell competition and cooperative tumorigenesis (of the kind examined in this study), both JAK–STAT and RAS–MAPK signalling pathways function to promote cell proliferation and tissue growth of the “winner” cells or tumorous cells. Additionally, during polarity-impairment-induced cell competition, JAK–STAT signalling has been shown to function in the “winner” cells adjacent to the polarity-deficient “loser” cells for the elimination of the polarity mutant cells [53]. Indeed, in isolation, STAT92E is capable of promoting a “winner” phenotype in cells if activated, and a “loser” fate if suppressed [61]. Current models suggest that polarity-impaired cells are eliminated through an intercellular feedback loop of cooperating JAK–STAT and JNK signalling pathways; JNK signalling is inherently upregulated in polarity-impaired cells, due to RHO1-WND (JNKKK) and EGR (TNF)-GRND (TNFR) signalling [10,11,14,62,63,64], which together with Yorkie (YKI, a co-transcriptional activator that is inhibited by the Hippo negative tissue growth control pathway) induces transcription and secretion of the Unpaired (UPD) family of ligands (Figure 8) [57,58]. These ligands activate the JAK–STAT signalling pathway in neighbouring *wild-type* cells via the receptor Domeless (DOME), and the pathway then upregulates compensatory proliferation, to replace the soon-to-be eliminated polarity-impaired cells, as well as promoting JNK signalling within the polarity-impaired cell via an unknown “competition signal” (Figure 8) [53]. Apart from the obvious questions regarding the nature of this “competition signal”, this model lacks an explanation for how JAK–STAT signalling is suppressed in the polarity-impaired cells. It is well established that UPD family ligands can signal in both a paracrine manner (such as during polarity-impairment-induced cell competition, promoting “winner” cell compensatory proliferation [53,58]) and an autocrine manner (such as during cooperative tumorigenesis driven by activated RAS and polarity-impairment, promoting tumor cell survival and proliferation leading to tumor overgrowth [58,65]). Therefore, based on our findings, and those of other groups regarding the requirement for JAK–STAT signalling in cell competition [15,53,58,61], we envisage that JAK–STAT regulators control the relative levels of JAK–STAT signalling in the *wild-type* winner cells versus the polarity-impaired loser cells. During cell competition, low levels/activity of PTP61F in the winner cells may facilitate elevated JAK–STAT signalling in these cells, while high levels/activity of PTP61F in the polarity-impaired cells may contribute to the downregulation of JAK–STAT signalling within these mutant cells (Figure 8). However, since the observed effect of PTP61F knockdown in the polarity-impaired cells was only small, whilst knockdown of STAT92E had a more robust effect on reducing *scrib* mutant clonal size, it is likely that other JAK–STAT regulators are also involved. Interestingly, we have also found a role for other established JAK–STAT pathway regulators acting in cooperation with *Ptp61F* during cell competition; namely *Socs44A* and *Socs36E*, which are thought to mediate the lysosomal degradation of Domeless, the *Drosophila* JAK–STAT pathway receptor and thereby downregulate pathway signalling (reviewed in [66,67]). We observed that both *Socs44A* and *Socs36E* knockdown, similar to *Ptp61F* knockdown, had a suppressive effect on the elimination of *scrib*-mutant clones, though it was a slightly larger difference (*Socs44A*/*Socs36E* knockdown clones resulting in a ~5% increase in tissue volume relative to only ~2% for *Ptp61F* knockdown clones) (Appendix A). Thus, multiple negative regulators of JAK–STAT signalling may act together to suppress JAK–STAT activity in loser cells during polarity-impaired cell competition (Figure 8). It is unclear, however, how these JAK–STAT regulatory genes might themselves be regulated in imaginal disc tissues and during cell competition. It has been demonstrated that *Ptp61F* expression can be repressed by JAK–STAT signalling in the developing testis [68], but, conversely, upregulated by JAK–STAT signalling in the embryo [52]. *Socs36E* is also thought to be upregulated by JAK–STAT signalling during embryogenesis and oogenesis [69], contributing to a negative feedback loop [70,71], but *Socs44A* expression is not regulated by JAK–STAT signalling [69]. How PTP61F (as well as the SOCS proteins) is regulated in *Drosophila* imaginal disc development and in tissues undergoing cell competition will be important to determine. Transcriptional and post-transcriptional regulation of *PTPN1* in mammalian tissues is, by contrast, relatively well understood (reviewed in [72]), and might provide a starting point for these investigations.

PTP61F is a known regulator of the phosphorylation-dependent RAS–MAPK signalling pathway [27,28], and we have demonstrated here a role for PTP61F in regulating RAS–MAPK signalling in an activated RAS-driven polarity-impaired cooperative tumorigenesis model. RAS–MAPK signalling has not been comprehensively explored in polarity-impairment-induced cell competition, but has recently been shown to be regulated by cell–cell interactions involving protein tyrosine phosphatase 10D (PTP10D) and Stranded at second (SAS) [73]. This is a recently identified cell competition regulatory mechanism that relies on physical interactions between SAS, located on the “winner” cells, and PTP10D, located on the “loser” cells [73]. PTP10D is a receptor-like or transmembrane PTP that is also capable of downregulating RAS–MAPK signalling [73]. In polarity-impaired “loser” cells deficient for this PTP10D–SAS interaction, it is believed the Epidermal Growth Factor Receptor (EGFR) is activated and promotes excess cell proliferation, due to RAS–MAPK pathway cooperation with co-opted JNK signalling [73]. Notably, other than PTP61F, PTP10D is the only PTP demonstrated to have a role during polarity-impairment-induced cell competition. Given the established role for each of these PTPs in downregulating RAS–MAPK signalling, is plausible that PTP61F might also contribute to the suppression of this pathway in polarity-impaired “loser” cells. It is not known whether the PTP10D–SAS system has any regulatory effect upon JAK–STAT signalling, but it is clear it has an overall greater effect upon relative fitness of the polarity-impaired cells than PTP61F does. However, since we have shown that STAT92E knockdown in *scrib* mutant clones has a strong effect on their size, it is likely that JAK–STAT activity in the polarity-impaired clones influences their survival. Although PTP61F may play only a small part in the regulation of JAK–STAT activity in polarity-impaired cells, this is likely due to redundant JAK–STAT inhibitory mechanisms, such as those involving other PTPs and the SOCS family members, SOCS44A and SOCS36E, as we have shown (Appendix A). It is also possible that additional cell competition regulatory mechanisms also contribute to polarity-impaired cell elimination, such as the “Flower code” [74,75], and the Toll signalling pathway [76]. Importantly, our results revealed a robust effect of *Ptp61F* up- or downregulation in the *wild-type* cells surrounding *scrib* mutant cells, consistent with the reported non-cell-autonomous role played by JAK–STAT signalling in polarity-impaired mutant cell elimination [53], and suggesting that the control of PTP61F expression levels in the surrounding *wild-type* cells, and thereby JAK–STAT signalling in these cells, may be an important regulatory mechanism in polarity-impaired cell competition. Moving forward, understanding how the various cell competition regulatory systems (e.g., SAS-PTP10D, “Flower code”, Toll signalling) control PTP61F (as well as SOCS44A and SOCS36E) expression, and JAK–STAT signalling during cell competition will be important to determine.

### 3.3. Human Orthologs of PTP61F in Tumorigenesis

We have demonstrated *Ptp61F* acts as a tumor suppressor within the context of *Ras85D^V12^*/*dlg1^RNAi^*-driven cooperative tumorigenesis, a robust *Drosophila* model of cancer development (reviewed in [4]). As mentioned, both mammalian orthologues of PTP61F, PTP1B and TCPTP (encoded by *PTPN1* and *PTPN2*, respectively), have roles in multiple cancers, though not necessarily as tumor suppressors (reviewed in [48,72]). Although PTP1B can act as a tumor suppressor, and its deletion in mice promotes increased development of acute leukemia with age [77], PTP1B predominantly functions as an oncoprotein, and its inhibition or deletion can attenuate oncogene-induced tumorigenesis. Indeed, PTP1B has been shown to function as an oncoprotein in breast [39,40], prostate [78,79], gastric [80], and colon [37] cancers, as well as squamous cell carcinoma [81]. In many cases, including breast cancers, PTP1B levels are increased and this correlates with advancement of the disease and worsened prognoses [39,40]. In mice, the deletion or inhibition of PTP1B attenuates the development of mammary tumors driven by mutant ErbB2 (HER-2) as well as their metastases, indicating that PTP1B plays an oncogenic role in the initiation of tumorigenesis [39,40,82]. The precise mechanisms by which increased PTP1B promotes tumorigenesis in ErbB2-driven breast cancers remain unknown, but this may be reliant on the promotion of RAS–MAPK signalling. Interestingly, PTP1B can serve as a positive regulator of RAS signalling by dephosphorylating and inactivating the RAS signalling suppressor docking protein 1 (DOK1, *Drosophila* orthologue Dok) [40,83,84].

By contrast, TCPTP is better understood as a tumor suppressor than an oncoprotein. Interestingly, TCPTP can exist as two variants derived from alternate mRNA splicing: A 48 kDa variant (TC48) that (like PTP1B) is targeted to the endoplasmic reticulum, and a 45 kDa variant (TC45) that shuttles between the nuclear and cytoplasmic environments [41,42,43,44]. As such, TCPTP has the capacity to attenuate tyrosine phosphorylation-dependent signalling in both the cytoplasm and nucleus. TCPTP shares multiple potentially oncogenic targets with PTP61F, including the EGFR-family member ErbB1 (also related to *Drosophila* EGFR), platelet-derived growth factor receptors (PDGFRs; related to *Drosophila* PVR signalling), insulin receptor (IR; related to *Drosophila* INR signalling), and multiple members of the JAK–STAT signalling pathway (JAK1/3 and STAT1/3/5/6, related to *Drosophila* Hopscotch (HOP) and STAT92E) (reviewed in [44,48]). TCPTP deficiency is thought to contribute to the progression of solid tumors and haemotological malignancies, including T cell acute lymphoblastic leukemia (T-ALL) [45,46], and some breast cancers [47]. The tumor-suppressive role of TCPTP in T-ALLs is attributed to its capacity to attenuate oncogenic protein tyrosine kinase signalling, including that mediated by JAK1 [46], and the aberrant fusion protein NUP214-ABL1 [45]. In mice, the conditional deletion of *Ptpn2* can also contribute to the development of skin cancers [85], as well as hepatocellular carcinomas [86,87], due to the promotion of STAT3 signalling. There is also evidence that deficiencies in TCPTP might contribute to tumorigenesis by promoting genome instability due to cell cycle checkpoint skipping [88,89]. Our findings with regard to PTP61F are consistent with the tumor-suppressive roles of TCPTP. Indeed, in unpublished studies, we have shown that although PTP1B, TC48, and TC45 can all attenuate the eye overgrowth associated with INR expression in *Drosophila*, only TC45 rescues the fecundity defects in *Ptp61F**^Δ/^**^Δ^* animals. Therefore, our findings in flies may provide important insights into how TCPTP deficiencies may lead to perturbations in cell competition and tumorigenesis in mammals.

It is currently unknown to what extent the various properties of PTP1B and TCPTP are conserved in PTP61F, besides some of those we have discussed above. In this study, we have revealed novel roles for PTP61F as a polarity impairment-induced cell competition regulator and a tumor suppressor in activated RAS-driven polarity-impaired (*Ras85D^V12^*/*dlg1^RNAi^*) tumorigenesis. These roles align with those that might be expected, particularly for TCPTP, and support the utility of *Drosophila melanogaster* for the examination of PTPs in cancer research. Furthermore, this work highlights the need for exploration of potential roles for PTP1B and TCPTP in mammalian cell competition. Cell competition as a mechanism for mammalian tissue homeostasis and tumor suppression is an emerging field (reviewed in [90]), and one with multiple parallels to observations from *Drosophila*, including the ability of cell polarity-impairment to induce the cell competition surveillance mechanisms [91,92]. Similar forays could be made into examining the role of PTPs in polarity-impaired cooperative tumorigenesis in mammals. Whilst the involvement of JAK–STAT signalling has not been comprehensively examined in polarity-impaired mouse models of cancer or in human cancer [93,94], cooperation between RAS–MAPK and JAK–STAT signalling has been noted in hepatocellular carcinoma [95], and more recently a conserved cooperation between RAS–MAPK and JAK–STAT signalling was observed in the development of radiation resistance in RAS-driven tumor models [96]. Polarity impairment and RAS oncogenes show strong cooperation in mouse cancer models [93,97], and elevated JAK–STAT signalling has been observed in at least one instance [93], but whether PTP1B and TCPTP are involved in mammalian polarity-impaired cancer models is currently unknown. As polarity impairment is a common event in human cancers, and occurs as an early event in cancer development [98,99,100,101], understanding the role that PTP1B and TCPTP play in the regulation of signalling pathways in polarity-impaired cancer models may provide new ways by which to target cancers at an early stage to enhance cell competition mechanisms and tumor suppressor functions.

## 4. Materials and Methods

### 4.1. Fly Stocks and General Husbandry

For a complete list of all fly stocks used in this study, please refer to Appendix A. For a list of complete genotypes, see Appendix A. All stocks and crosses were raised and undertaken on a standard cornmeal/molasses/yeast medium within temperature-controlled incubators at 25 °C, unless otherwise indicated.

### 4.2. Imaging and Sample Preparation

L3 eye-antennal or wing imaginal discs were dissected in 1× phosphate-buffered saline (PBS, Amresco (Radnor, PA, USA) #E703) and fixed in 4% paraformaldehyde (Alfa Aesar (Haverhill, MA, USA) #43368) in PBS with 0.1% Triton X-100 (PBST) for 15–30 min. Samples were then washed in 0.1–0.3% PBST (0.1% or 0.3% Triton X-100). If no immunohistochemical staining was required, samples were then incubated in DAPI solution (for DNA detection, DAPI solution prepared at 1 μg/mL, used at 1:1000, Sigma-Aldrich (St. Louis, MO, USA) #D9542) and/or phalloidin-tetramethylrhodamine isothiocyanate-Rhodamine solution (for F-actin detection, stock prepared at 0.3 mM, used at 1:1000, Sigma-Aldrich #P1951) in 0.1% Triton X-100. After DNA/F-actin staining, samples were mounted in 80% glycerol. Samples were analysed via confocal microscopy using either a Leica TCS SP5 or a Zeiss LSM 780. Images were processed, analysed, and assembled using some combination of LAS AF Lite (Leica (Wetzlar, Germany)), Zen 2012 (Zeiss (Oberkochen, Germany)), Fiji [102], Photoshop 5.1/2019/2020 (Adobe (San Jose, CA, USA)), and Imaris v.7 (Bitplane (Zürich, Switzerland)).

### 4.3. Twin-Clone Generation and Quantification

Larvae were heat shocked at 37 °C for 1 h approximately 72 h after egg laying. L3 imaginal discs were then dissected and prepared as described, and clearly distinct twin-clones had their sizes were measured across Z-stacks using Fiji [102]. Statistical analyses comparing the size ratio between the *wild-type/wild-type* twin-clones and the *wild-type/Ptp61F^Δ/Δ^* twin-clones was performed using Student’s unpaired *t*-test in Prism 7/8/9 (GraphPad (San Diego, CA, USA)). Technique adapted from Froldi et al. [50].

### 4.4. MARCM and Reverse MARCM

Crosses were set up and left overnight at room temperature. Adults were then turned daily into new vials and allowed to lay for ~24 h at 25 °C. L3 animals were then collected after ~144 h (6 days after the egg laying period). Samples were collected and prepared as described across multiple days to be similarly aged, and then stored at 4 °C in 80% glycerol as necessary prior to mounting.

### 4.5. Clone Volume Quantification and Statistical Analyses

Samples were imaged by adjusting the gain levels to just below saturation for the GFP and DAPI detection channels, and Z-stacks were acquired using an LSM 780 confocal microscope with Zen 2012 (Zeiss). Clone and whole tissue volume measurements were performed using Imaris v.7 (Bitplane). First, a surface was generated and used to mask and remove extraneous tissue from the sample. Then, the program automatically set threshold values for the GFP and DAPI channels, and surfaces were generated to identify the volumes of the whole tissue (DAPI positive) and clones (GFP positive). Finally, the GFP-positive/DAPI-positive tissue volume ratio was obtained, to give the proportion of each sample populated by clonal tissue. Data were collated using Excel (Microsoft) and statistically analysed using Prism 7/8/9 (GraphPad). The particular statistical tests employed are detailed in the respective figure legends. Note, for the samples in Figure 3, these experiments were performed under the same conditions and soon afterwards those using *Ptp61F^RNAi^* discussed in Figure 2. Therefore, we utilised the same samples as controls.

### 4.6. Statistical Analyses of Signal Intensity and the eq Expression Domain Area

Z-stack images of the eye region of eye-antennal imaginal discs derived from L3 animals were obtained using identical microscope settings and used to generate maximum intensity projection images. Average pixel intensity of 10×STAT92E-GFP expression was measured using histogram tool in Photoshop 5.1 (Adobe), using an area of 400–2500 pixels located just posterior of the morphogenetic furrow. Average pixel intensity both within and without the *eq* expression domain was measured (*n* = ~10 for each sample) and expressed as the ratio of *eq* expression domain pixel intensity to *wild-type* tissue pixel intensity. Data were collated using Excel (Microsoft), and analysed using Prism 7/8 (GraphPad).

### 4.7. Western Blotting, Analyses, and Antibodies

Samples were prepared from ~20 eye-antennal imaginal discs and homogenized in lysis buffer (0.1 M Tris-HCl (pH 6.8), 2% SDS, 5 mM EDTA, 5 mM DTT) containing cOmplete Protease Inhibitor Cocktail (Roche (Basel, Switzerland) #11697498001), and 1 mM Na_3_(VO)_4_ and 5 mM NaF_2_ as phosphatase inhibitors. Samples were boiled for 5 min in 1 × Laemmli SDS buffer plus 5% (*v*/*v*) *β*-mercaptoethanol and proteins resolved by SDS-PAGE (10%). Gels were run at 100 V, and Precision Plus Protein Standards (Bio-Rad (Hercules, CA, USA)) were used as a ladder. Western blotting was performed with a Trans-Blot Turbo Transfer pack (Bio-Rad, #170-4156, #170-4158) and Immun-Blot PVDF membrane (Bio-Rad, #1620177). Membranes were blocked in 5% skim milk blocking solution and washed with TBST, then placed in primary antibody solutions overnight. Primary antibodies used were: mouse anti-Phosphotyrosine, clone 4G10 (Merck & Co. (Kenilworth, NJ, USA) #05-321), rabbit anti-phospho-p44/42 MAPK (Erk1/2) (Thr202/Tyr204) (a.k.a. pERK, Cell Signaling Technology (Danvers, MA, USA) #9101), rabbit anti-ERK 2 (D-2) (a.k.a. ERK, Santa Cruz Biotechnology (Dallas, TX, USA) #sc-1647), mouse anti-α-Tubulin (DM1A) (Cell Signaling Technology #3873), and mouse anti-β-Galactosidase (Sigma-Aldrich #G4644). Blots were then washed in TBST and incubated in secondary antibodies for at least 60 min. Secondary antibodies used include goat anti-mouse IgM/G/A (H+L) F(ab’)2 HRP conjugate (Millipore (Burlington, MA, USA) #AQ502P) and goat-anti-rabbit IgG F(ab’)2 HRP conjugate (Millipore #AQ132P). Proteins were visualised using Clarity Western ECL Substrate (Bio-Rad #170-5061). Band intensity was measured using Fiji [102], and data analysed using Excel (Microsoft).

### 4.8. RNA Extraction, cDNA Synthesis, and qRT-PCR

RNAi lines were crossed to *eyFLP; Actin>CD2>GAL4, UAS-GFP*, and raised at 25 °C. L3 eye-antennal discs (*n* > 10 per for each genotype) were obtained as described for immunofluorescence above, with RNA extraction and cDNA synthesis then performed as previously described [103]. qRT-PCR was performed using a Power SYBR Green PCR Master Mix (Applied Biosystems (Waltham, MA, USA) #4367659) on a QuantStudio 12K Flex Real-Time PCR System (Applied Biosystems). The data were normalised to expression of the housekeeping genes *Gapdh2* and *Rpl32*. The primer sequences used are as follows: *Stat92E*, forward 5′-TGCTCCGTTTCTCCGACAG-3′ and reverse 5′-CTAGCATGGTGACCAGTCC-3′; *Ptp61F*, forward 5′-GTGCGGCGATGGTTCAAATTA-3′ and reverse 5′-CTTAAGGAATGCGTTCGGCG-3′; *Gapdh2*, forward 5′-GCAAGCAAGCCGATAGATAAACA-3′ and reverse 5′-CGTTGGCGCCCTTATCAATG-3′; *Rpl32*, forward 5′-CCAGTCGGATCGATATGCTAA-3′ and reverse 5′-GTTCGATCCGTAACCGATGT-3′.

## Figures and Tables

**Figure 1 ijms-22-12732-f001:**
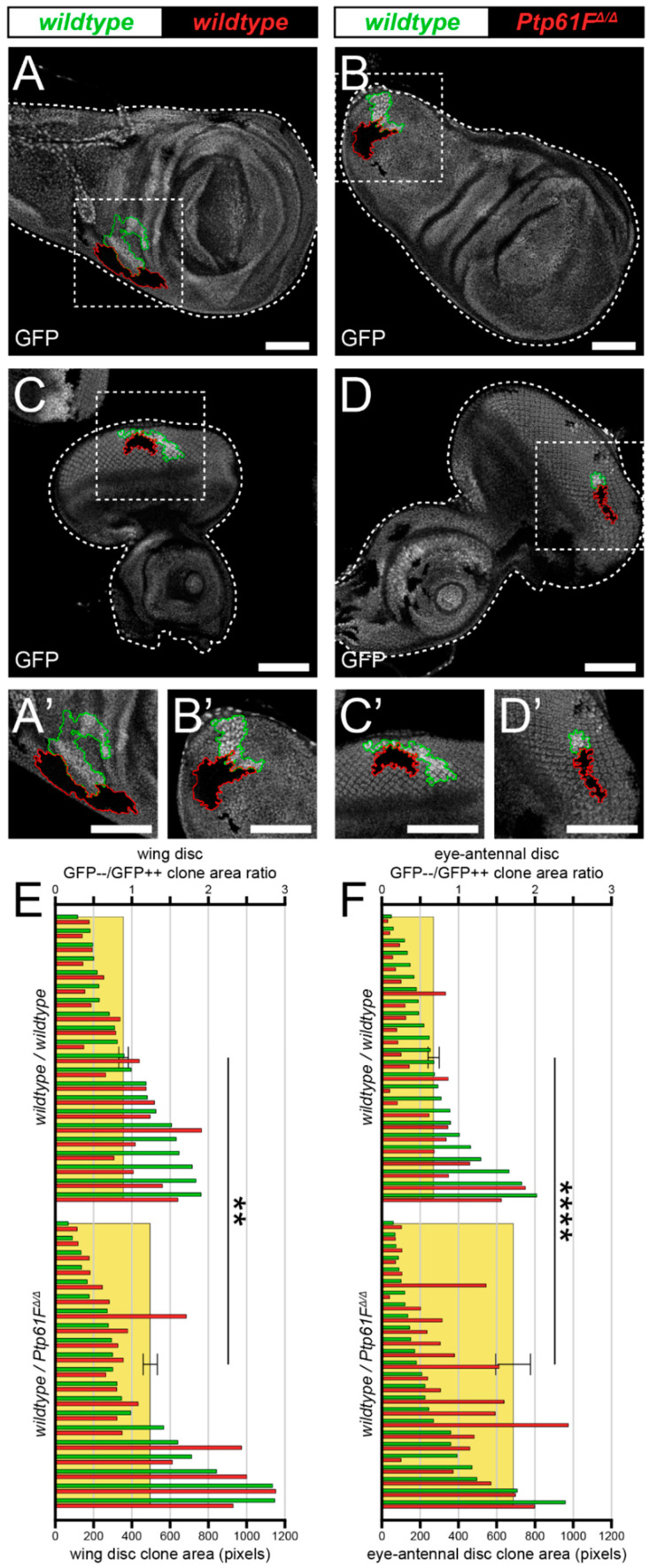
*Ptp61F* loss enhances epithelial clone relative fitness. (**A**–**D**) Confocal images of L3 imaginal tissues of the indicated genotypes (boxes at top, white indicates GFP-positive clones, black indicates remaining GFP-negative tissue) taken from animals where twin-clones were generated. The twins were either GFP-double positive (outlined in green) or GFP negative (outlined in red), with all other cells being GFP-single positive. (**A**,**B**) Twin clone analysis in wing imaginal discs. (**A**,**A’**) In wing imaginal discs, the GFP-double-positive clones in *wild-type/wild-type* twins are slightly larger (twin-clone size ratio $\bar{x}$ = 0.895 ± 0.091). (**B**,**B’**) The reverse is true for wing disc *wild-type/Ptp61F^Δ/Δ^* twin-clones, with the GFP-negative mutant clones being generally larger (twin-clone size ratio $\bar{x}$ = 1.244 ± 0.093). (**C**–**D**) Twin clone analysis in eye-antennal imaginal discs. (**C**,**C’**) In eye-antennal imaginal discs, the GFP-double-positive clones in *wild-type/wild-type* twin-clones are again generally larger (twin-clone size ratio $\bar{x}$ = 0.680 ± 0.072). (**D**,**D’**) Similarly, eye-antennal *wild-type/Ptp61F^Δ/Δ^* twin-clones have generally larger GFP-negative mutant clones (twin-clone size ratio $\bar{x}$ = 1.712 ± 0.228). (**E**) Quantification of L3 wing imaginal disc clone size profiles from *wild-type/wild-type* and *wild-type/Ptp61F^Δ/Δ^* twin-clone pairs. Green bars indicate the GFP-double-positive clone of the twin-clone pair, and red bars indicate the GFP-negative clone, and use the lower *x*-axis. Yellow rectangles indicate the average GFP-negative/GFP-double-positive clone area ratios, and use the upper *x*-axis, showing that *Ptp61F^Δ/Δ^* clones are significantly larger that the *wild-type* twin-clones (Student’s *t*-test, d.f. = 39, *t* = 3.169, *p* < 0.01). (**F**) Quantification of L3 eye-antennal imaginal disc clone size profiles from *wild-type/wild-type* and *wild-type/Ptp61F^Δ/Δ^* twin-clone pairs. Green bars indicate the GFP-double-positive clone of the twin-clone pair, and red bars indicate the GFP-negative clone, and use the lower *x*-axis. Yellow rectangles indicate the average GFP-negative/GFP-double-positive clone area ratios, and use the upper *x*-axis, showing that *Ptp61F^Δ/Δ^* clones are significantly larger that the *wild-type* twin-clones (Student’s *t*-test, d.f. = 47, *t* = 4.272, *p* < 0.0001). ** = *p* < 0.01, **** = *p* < 0.0001. Error bars = S.E.M. Note that clones were observed in all regions of both the eye-antennal and wing disc tissues, but that the figures show representative clones with GFP-negative/GFP-double-positive clone area ratios close to the average. Confocal microscopy images are single planes. Boxes in (**A**–**D**) are represented in (**A’**–**D’**), and dotted lines outline the tissue. Scale bars = 100 μm.

**Figure 2 ijms-22-12732-f002:**
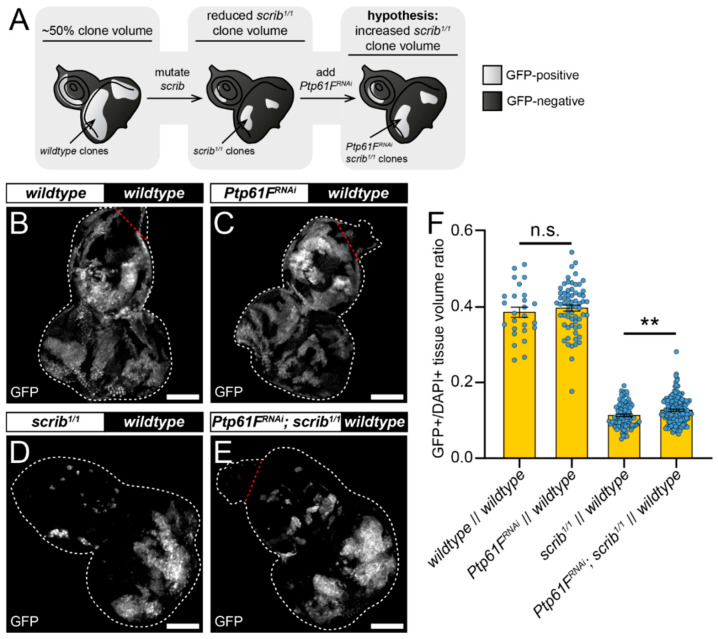
*Ptp61F* contributes to *scrib*-mutant clone elimination. (**A**) Diagram of our experimental process and hypothesis. (**B**–**E**) Confocal images of L3 eye-antennal imaginal discs of the indicated genotypes (boxes at top, white indicates GFP-positive clones, black indicates remaining GFP-negative tissue) taken from animals where clones were generated via MARCM to express transgenes in *scrib*-mutant, GFP-positive cells. (**B**,**C**) When GFP-positive tissue is *wild-type* (**B**) it makes up ~40% of the tissue (*n* = 26, $\bar{x}$ = 0.386 ± 0.013), and *Ptp61F^RNAi^* expression (**C**) does not significantly alter the contribution of the clones to the tissue (*n* = 64, $\bar{x}$ = 0.397 ± 0.008) (Student’s *t*-test, d.f. = 88, *t* = 0.7263, *p* > 0.05). (**D**,**E**) *scrib*-mutant clones (**D**) make up only ~11% of the tissue (*n* = 88, $\bar{x}$ = 0.113 ± 0.003), but *Ptp61F^RNAi^* expressed in *scrib*-mutant clones (**E**) leads to a small, but statistically significant increase in clonal volume to ~13% (*n* = 144, $\bar{x}$ = 0.127 ± 0.003) (Student’s *t*-test, d.f. = 230, *t* = 2.829, *p* < 0.01). (**F**) Quantification of the clone tissue volume contributions, as measured by the ratio of GFP-positive tissue to DAPI-positive tissue, showing that *Ptp61F* knockdown significantly increases the size of *scrib*-mutant clones. Note that *Dcr-2* is also expressed wherever GFP is expressed. ** = *p* < 0.01. Error bars = S.E.M. Confocal microscopy images are maximum intensity projections. White dotted lines outline the tissue, red dotted lines indicate tissue excluded from quantification for consistency. Scale bars = 100 μm.

**Figure 3 ijms-22-12732-f003:**
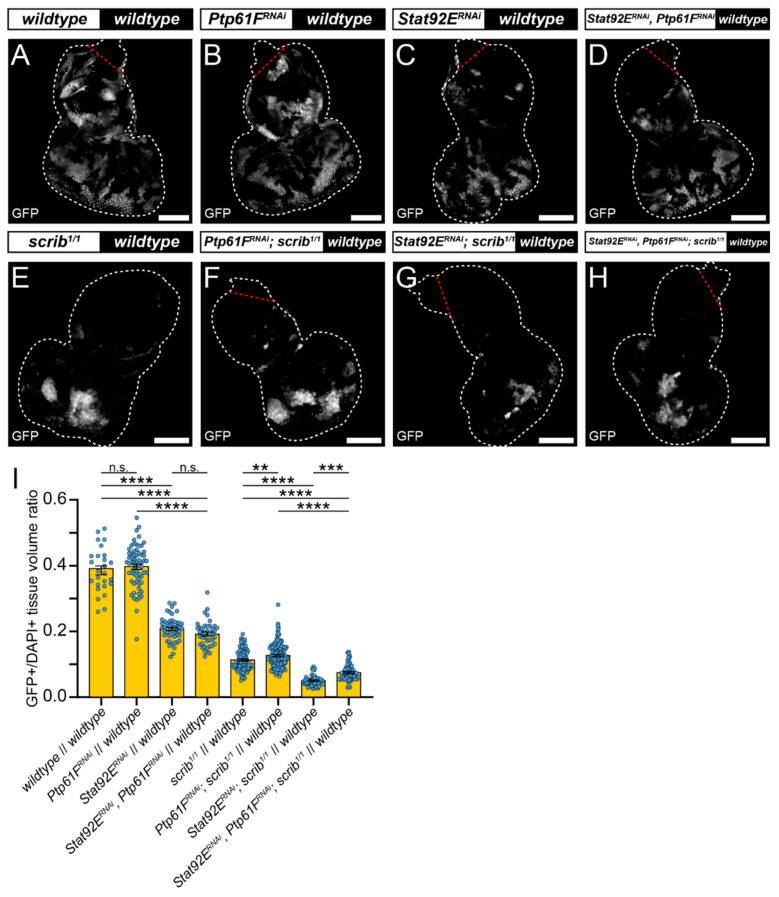
*Stat92E* is required for rescue of *scrib*-mutant clone size by *Ptp61F* knockdown. (**A**–**H**) Confocal images of L3 eye-antennal imaginal discs of the indicated genotypes (boxes at top, white indicates GFP-positive clones, black indicates remaining GFP-negative tissue) taken from animals where clones were generated via MARCM to express transgenes in GFP-positive cells. (**A**–**D**) *Wild-type* clones (**A**) and *Ptp61F^RNAi^*-expressing (**B**) clones appear largely the same (for statistics see Figure 2). *Stat92E^RNAi^* expression in a *wild-type* background ((**C**); *n* = 48, $\bar{x}$ = 0.208 ± 0.005) leads to significantly smaller clones compared to the *wild-type* control (one-way ANOVA ((**F**) (3,175) = 193.3, *p* < 0.0001), with Tukey’s multiple comparisons (*p* < 0.0001)), and combining *Stat92E* and *Ptp61F* knockdown in a *wild-type* background ((**D**); *n* = 41, $\bar{x}$ = 0.192 ± 0.006) does not result in significantly different clone sizes to *Stat92E* knockdown alone (one-way ANOVA ((**F**) (3,175) = 193.3, *p* > 0.05), with Tukey’s multiple comparisons (*p* > 0.05)), but does significantly reduce the average clone size relative to *Ptp61F^RNAi^* clones (one-way ANOVA ((**F**) (3,175) = 193.3, *p* > 0.05), with Tukey’s multiple comparisons (*p* > 0.05)). (**E**–**H**) Clones homozygous mutant for *scrib* (**E**) have their reduced volume somewhat rescued by *Ptp61F* knockdown (**F**) (for statistics see Figure 2). Knockdown of *Stat92E* in a *scrib*-mutant background ((**G**); *n* = 51, $\bar{x}$ = 0.050 ± 0.002) leads to clones that are significantly smaller in their contribution to the tissue volume than the *scrib*-mutant control (one-way ANOVA ((**F**) (3,342) = 97.72, *p* < 0.0001), with Tukey’s multiple comparisons (*p* < 0.0001)). Simultaneous knockdown of *Stat92E* and *Ptp61F* in *scrib^1/1^* clones ((**H**); *n* = 63, $\bar{x}$ = 0.075 ± 0.003) led to a statistically significant increase in clone volume relative to *scrib*-mutant, *Stat92E* knockdown clones (one-way ANOVA ((**F**) (3,342) = 97.72, *p* < 0.0001), with Tukey’s multiple comparisons (*p* < 0.001)), and also resulted in a statistically significant decrease in clone volume compared to *scrib*-mutant, *Ptp61F^RNAi^*-expressing clones (one-way ANOVA ((**F**) (3,342) = 97.72, p < 0.0001), with Tukey’s multiple comparisons (*p* < 0.0001)). (**I**) Quantification of the clone tissue volume contributions, as measured by the ratio of GFP-positive tissue to DAPI-positive tissue, showing that simultaneous *Ptp61F* and *Stat92E* knockdown significantly increases the size of *scrib*-mutant clones compared to *Stat92E* knockdown alone in *scrib*-mutant clones. Note that some sample sets here are taken from Figure 2, as the experiments were performed under the same conditions and soon afterwards allowing them to be utilised as controls. Note that *Dcr-2* is also expressed wherever GFP is expressed. ** = *p* < 0.01, *** = *p* < 0.001, **** = *p* < 0.0001. Error bars = S.E.M. Confocal microscopy images are single planes. White dotted lines outline the tissue, red dotted lines indicate tissue excluded from quantification for consistency. Scale bars = 100 μm.

**Figure 4 ijms-22-12732-f004:**
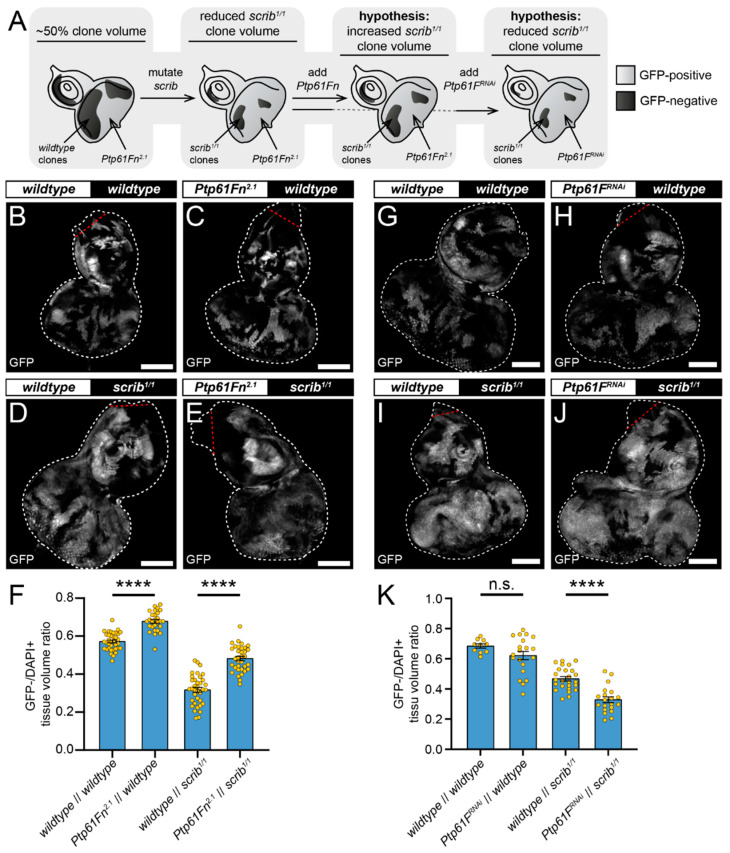
Nuclear-localised PTP61F expression reduces the relative fitness of *wild-type* cells during polarity-impaired cell competition. (**A**) Diagram of our experimental process and hypothesis. (**B**–**E**,**G**–**J**) Confocal images of L3 eye-antennal imaginal discs of the indicated genotypes (boxes at top, white indicates GFP-positive clones, black indicates remaining GFP-negative tissue) taken from animals where clones had been generated via revMARCM to express transgenes in cells that are otherwise *wild-type*, alongside cells that are GFP-negative and *scrib*-mutant. (**B**,**C**) When GFP-negative tissue is *wild-type* it makes up ~60% of the whole tissue when GFP-positive clones are *wild-type* ((**B**); *n* = 36, $\bar{x}$ = 0.572 ± 0.008), but *Ptp61Fn^2.1^* expression leads to the GFP-negative tissue making up ~70% of the total tissue volume ((**C**); *n* = 27, $\bar{x}$ = 0.679 ± 0.010), a statistically significant increase (Student’s *t*-test, d.f. = 61, *t* = 8.456, *p* < 0.0001). (**D**,**E**) When GFP-negative tissue is homozygous mutant for *scrib^1^*, it only contributes to ~30% of the whole tissue when surrounded by *wild-type* clones ((**D**); *n* = 36, $\bar{x}$ = 0.317 ± 0.013), but *Ptp61Fn^2.1^* expression significantly increases the *scrib^1/1^* GFP-negative tissue contribution to ~50% of the total volume ((**E**); *n* = 34, $\bar{x}$ = 0.482 ± 0.011), also a statistically significant result (Student’s *t*-test, d.f. = 68, *t* = 9.331, *p* < 0.0001). (**F**) Quantification of the clone tissue volume contributions in (**B**–**E**), as measured by the ratio of GFP-negative tissue to DAPI-positive tissue. (**G**,**H**) When the tissue is entirely functionally *wild-type* GFP-positive clones make up ~60% of the tissue ((**G**); *n* = 10, $\bar{x}$ = 0.685 ± 0.014), and when *Ptp61F^RNAi^* is expressed adjacent to *wild-type* clones ((**H**); *n* = 20, $\bar{x}$ = 0.621 ± 0.028), there is no significant difference in contributing volumes (Student’s *t*-test, d.f. = 28, *t* = 1.565, *p* > 0.05). (**I**,**J**) Contrastingly, when *wild-type* GFP-positive clones are adjacent to *scrib*-mutant clones, the mutant tissue makes up ~50% of the tissue ((**I**); *n* = 25, $\bar{x}$ = 0.467 ± 0.015), but when *Ptp61F^RNAi^* is expressed adjacent to *scrib*-mutant clones ((**J**); *n* = 20, $\bar{x}$ = 0.328 ± 0.020), the mutant clone sizes are significantly reduced to ~33% (Student’s *t*-test, d.f. = 43, *t* = 5.730, *p* < 0.0001). (**K**) Quantification of the clone tissue volume contributions in (**G**–**J**), as measured by the ratio of GFP-negative tissue to DAPI-positive tissue. Note that *Dcr-2* is also expressed wherever GFP is expressed in (**G**–**J**). **** = *p* < 0.0001. Error bars = S.E.M. Confocal microscopy images are single planes. White dotted lines outline the tissue, red dotted lines indicate tissue excluded from quantification for consistency. Scale bars = 100 μm (**B**–**E**,**G**–**J**).

**Figure 5 ijms-22-12732-f005:**
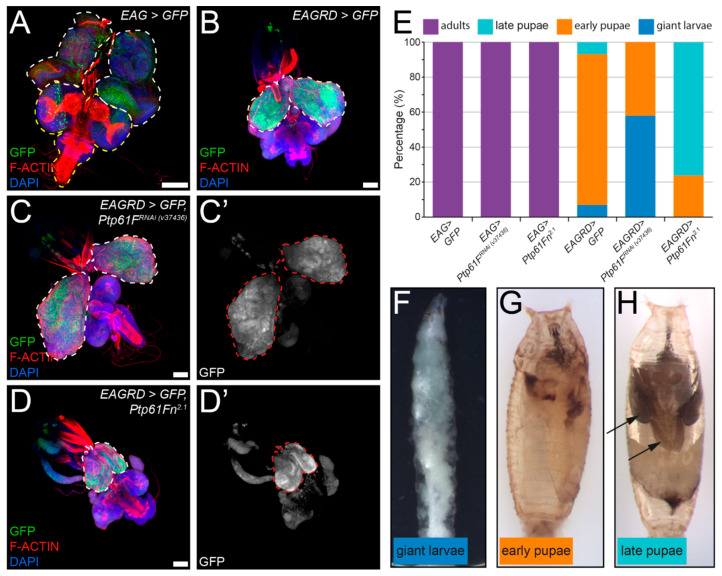
Activated RAS85D-driven polarity-impaired tumorigenesis is modified by PTP61F levels in L3 eye-antennal epithelial tissue. (**A**–**D**) Confocal images of eye-antennal/epithelial-brain complexes of the indicated genotypes taken from L3 animals (generated using the *eyFLP; Act >> GAL4* (*EAG*) system, also expressing *GFP*, *Ras85D^V12^* and RNAi against *dlg1* (*EAGRD*)). Tissues are stained with phalloidin to detect F-actin and DAPI to detect DNA. (**A**) *EAG > GFP* tissue shows the regular morphology of the eye-antennal discs (outlined in white) and the optic lobes and ventral nerve cord of the larval brain (outlined in yellow). (**B**) *EAGRD > GFP* tissue shows tissue overgrowth and perturbed cell morphology. (**C**,**C’**) *EAGRD*-driven tissue overgrowth is enhanced by co-expression of RNAi against *Ptp61F*. (**D**,**D’**) *EAGRD*-driven tissue overgrowth is suppressed by overexpression of *Ptp61Fn^2.1^*. (**E**) Quantification of the effect of *EAG*/*EAGRD* and alteration of *Ptp61F* levels on development. While EAG-driven expression of *GFP*, *Ptp61F^RNAi^*, or *Ptp61Fn^2.1^* does not appear to have any effect on development, *EAGRD*-driven expression of *GFP* in the control results in developmental arrest in most animals (~90%) at the early pupal stage, while ~5% of animals stall as overgrown (giant) larvae and ~5% reach the late pupal stage. *EAGRD*-driven expression of *Ptp61F^RNAi^* further stalled development, with no late-stage pupae being observed, and only ~40% reaching the early pupal stage, while *EAGRD*-driven expression of *Ptp61Fn^2.1^* somewhat rescued development, with ~80% of animals reaching the late pupal stage, though none successfully eclosed. (**F**–**H**) Examples of how the developmental phenotypes were graded. The distinction between early and late pupae relied on the observation of adult structures such as wings and legs (arrows). Confocal microscopy images are maximum intensity projections. White dotted lines outline the eye-antennal imaginal discs, and yellow dotted lines outline the brain structures where relevant. Scale bars = 100 μm.

**Figure 6 ijms-22-12732-f006:**
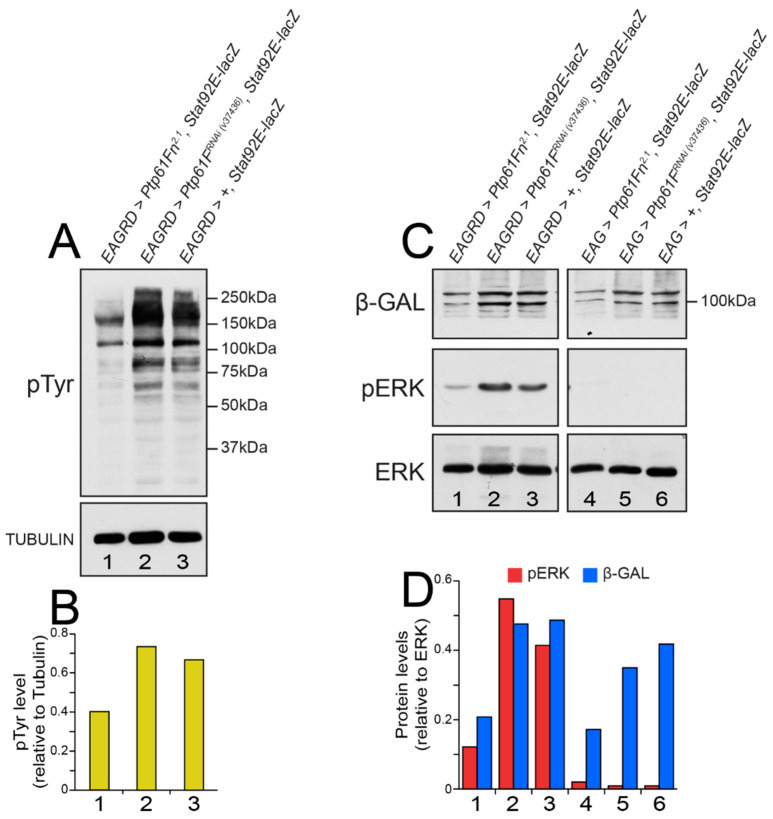
PTP61F represses tyrosine phosphorylation and JAK–STAT activity in activated RAS85D-driven polarity-impaired tumors. (**A**) Western blotting for the presence of tyrosine phosphorylated (pTyr) proteins and Tubulin protein levels in lysates from L3 eye-antennal imaginal disc tissues. *Ptp61F^RNAi^* or *Ptp61Fn^2.1^* was expressed throughout the eye-antennal tissue using the *EAGRD* system. (**B**) Quantification of pTyr levels. Relative to the Tubulin loading control, pTyr levels are higher in *EAGRD* samples expressing *Ptp61F^RNAi^*, and lower in those expressing *Ptp61Fn^2.1^*, relative to the control. (**C**) Western blotting for β-gal (STAT92E-lacZ), phosphorylated and activated ERK (pERK), and ERK protein levels in lysates from L3 eye-antennal imaginal disc tissues. *Ptp61F^RNAi^* or *Ptp61Fn^2.1^* was expressed throughout the eye-antennal tissue using the *EAG* or *EAGRD* systems. (**D**) Quantification of pERK and β-gal levels. While pERK is essentially undetectable in the *EAG* controls (relative to total ERK as a loading control), it is upregulated in *EAGRD* samples, and even more so upon *Ptp61F* knockdown, while being reduced upon PTP61Fn overexpression. JAK-STAT activity (as assessed using the STAT92E-lacZ reporter and measuring β-gal levels) was reduced upon *Ptp61Fn* expression, but *Ptp61F* knockdown had no notable effect. Note that in both (**B**,**D**), the numbered columns correspond to the numbered lanes in (**A**,**C**), respectively.

**Figure 7 ijms-22-12732-f007:**
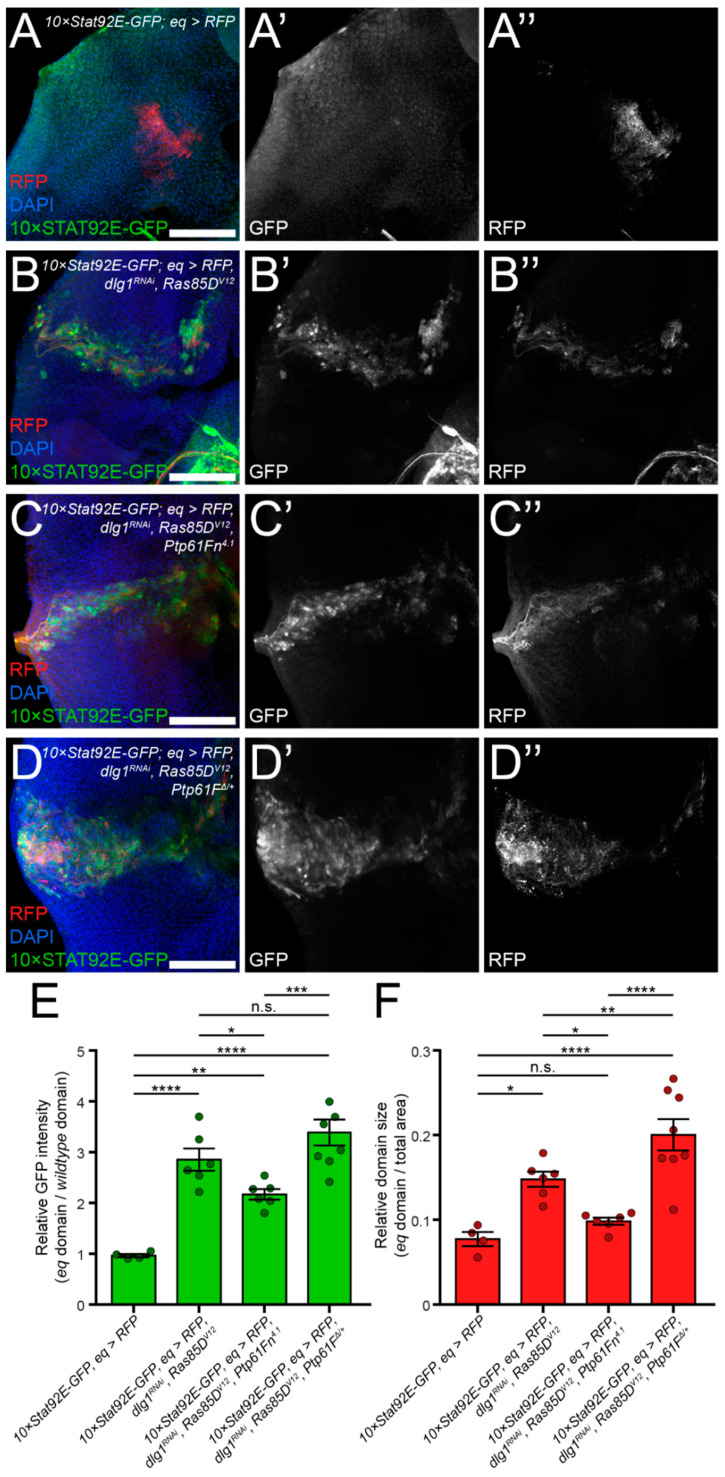
PTP61F represses JAK–STAT signalling pathway activity in activated RAS85D-driven polarity-impaired tumorigenic tissue. (**A**–**D**) Confocal images of L3 eye imaginal disc epithelial tissue containing the *10×Stat92E-GFP* reporter of JAK–STAT signalling pathway activity, the *eq-GAL4* driver transgene and *UAS-RFP* expression to mark the *eq* expression domain, was examined for expression levels of STAT92E-GFP and also stained with DAPI to mark DNA. (**B**–**D**) *Ras85D^V12^* and RNAi against *dlg1* are also expressed to generate tumorigenic tissue within the *eq* domain. (**A**,**A**,**A’’**) *eq* > *RFP* control discs demonstrate basal levels of STAT92E-GFP expression, a readout of JAK–STAT pathway activity. (**B**,**B’**,**B’’**) *eq > RFP, Ras85D^V12^, dlg1^RNAi^* discs possess an enlarged *eq* expression domain and elevated levels of STAT92E-GFP. (**C**,**C’**,**C’’**) *eq > RFP, Ras85D^V12^, dlg1^RNAi^* discs also expressing *Ptp61Fn^4.1^* show a reduction in the *eq* expression domain area, as well as a slight reduction in STAT92E-GFP level. (**D**,**D’**,**D’’**) *eq > RFP, Ras85D^V12^, dlg1^RNAi^* discs that are heterozygous mutant for *Ptp61F* (*Ptp61F**^Δ/+^*) show further expansion of the *eq* expression domain, but only a non-significant increase in STAT92E-GFP expression. (**E**) Quantification of the STAT92E-GFP pixel intensity within the *eq* expression domain relative to surrounding *wild-type* tissue, showing that overexpression of PTP61F reduces STAT92E-GFP expression. (**F**) Quantification of the *eq* domain area relative to the total eye disc area. *Ptp61Fn^4.1^* expression decreases the size of the *eq* expression domain, whilst heterozygosity for *Ptp61F* increases the *eq* domain region, relative to the *eq > RFP, Ras85D^V12^, dlg1^RNAi^* control. Statistical tests used were one-way ANOVAs with Newman–Keuls multiple comparisons tests. * = *p* < 0.05, ** = *p* < 0.01, *** = *p* < 0.001, **** = *p* < 0.0001. Error bars = S.E.M. Confocal microscopy images are maximum intensity projections. Scale bars = 50 µm.

**Figure 8 ijms-22-12732-f008:**
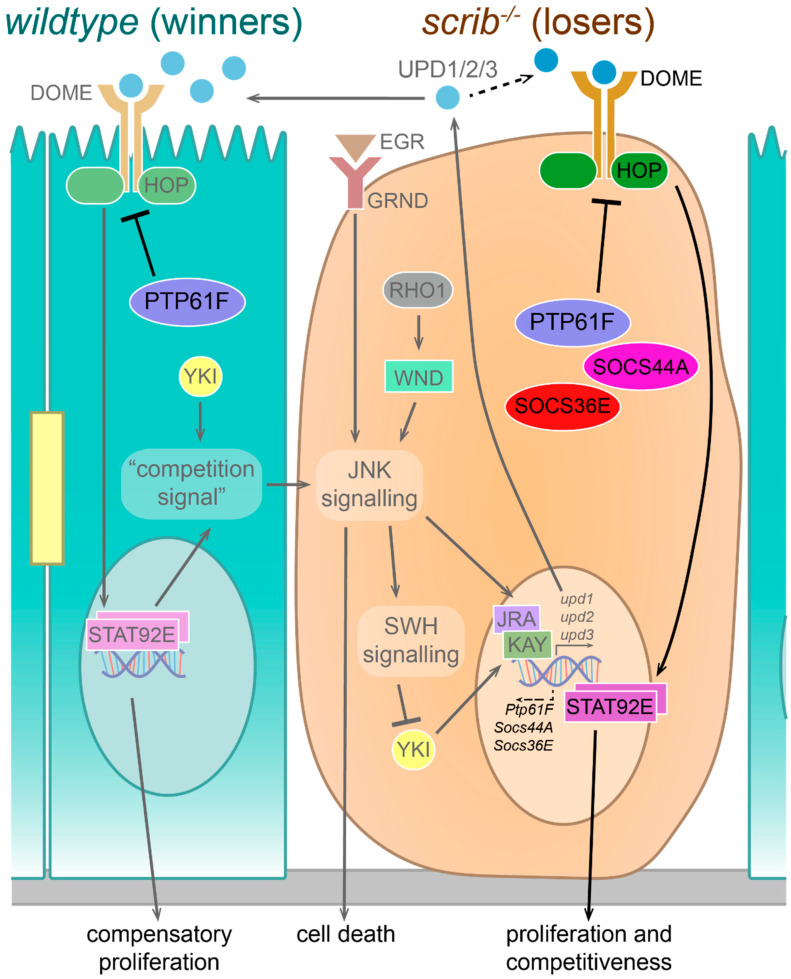
A model for PTP61F and JAK–STAT signalling during polarity-deficient cell competition. During polarity-deficient cell competition, the current model places the JAK–STAT pathway as having a role exclusively in the *wild-type* winner cells. JAK–STAT pathway activating ligands, the UPD molecules, are produced and secreted by polarity-deficient loser cells due to YKI activity and JNK signalling. JNK signalling itself is thought to be activated by RHO1-WND signalling, EGR-GRND signalling, and by a “competition signal” originating in the *wild-type* cells. This signal appears to lie downstream of JAK–STAT signalling, completing a cyclic process that culminates in the death and elimination of the polarity-deficient cells and the compensatory proliferation and persistence of the *wild-type* cells. However, we have identified a new role for JAK–STAT signalling within polarity-deficient cells. PTP61F, SOCS44A, and SOCS36E are all capable of downregulating the activity of the JAK–STAT signalling, and their knockdown slightly increases polarity-deficient cell survival, whereas knockdown of STAT92E has a strong effect on reducing polarity-deficient cell survival. This suggests JAK–STAT signalling is active within the polarity-deficient cells, contributing to their relative fitness. Whether JAK–STAT signalling is activated by autocrine UPD ligand signalling is unclear. Additionally, PTP61F also has an important non-cell-autonomous role in the *wild-type* cells in influencing the elimination of polarity-impaired cells, presumably by negatively regulating JAK–STAT signalling. Previously known data are shown in muted colours, while our findings are shown in bolder colours. Abbreviations: UPD = Unpaired, DOME = Domeless, HOP = Hopscotch, EGR = Eiger, GRND = Grindelwald, YKI = Yorkie, PTP61F = protein tyrosine phosphatase 61F, STAT92E = signal-transducer and activator of transcription protein at 92E, SOCS44A = suppressor of cytokine signalling at 44A, SOCS36E = suppressor of cytokine signalling at 36E, WND = Wallenda, JNK = c-Jun N-terminal Kinase, SWH = Salvador-Warts-Hippo, JRA = Jun-related antigen, KAY = Kayak.

## Data Availability

Not applicable.

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
