# Peer review of "PTP61F Mediates Cell Competition and Mitigates Tumorigenesis"

_ijms, 2021, doi:10.3390/ijms222312732_

Round 1

Reviewer 1 Report

In this study, La Marca et al. describe their characterization of the protein phosphatase PTP61F in a Drosophila cell competition and tumorigenesis model. The authors use well established polarity-impaired and/or Ras activated strains and characterize the effects of gain or loss of PTP61F on cell behaviors, mainly through the evaluation of clonal size and cell competition.   Overall, the authors do a very nice job explaining a complex set of assays. Most of their experiments involve generating marked cell populations of different genotypes and assessing cell autonomous and non-autonomous effects on clone size. These results are then quantified to determine significance. Their findings are convincing, and they shed light on a previously unknown role for PTP61F in these interactions. While the results are not surprising, given the known effects in other contexts, these experiments contribute more to our understanding of signaling crosstalk during tumorigenesis.    Specific comments: 1. Many of the figure legends (2, 3, 4) mistakenly say “wing imaginal discs” while eye discs are shown. 2. The authors initially examine clones in wing discs, then switch entirely to eyes. Was there a rationale for this? The two clones shown in the wing disc appear to be in the notum and hinge precursors. Were clones observed in the pouch domain? 3. Fig. 1E, F – the legend describes yellow bars. It would be clearer to say yellow rectangles. I presume the area of the rectangles reflect the size, rather than just location of the end of the rectangle?  4. The authors in Fig. 1show that loss of PTP61F gives clones a growth advantage over neighboring wildtype cells and use this fact as the starting point for their studies. This effect is not seen with the PTP61F RNAi in Fig. 2 & 3. The authors should point out that this effect is weaker and why they opted to use RNAi. 5. Do the authors use Gal4 titration controls? In cases where numerous Gal4-driven transgenes are used and effects are compared, want to make sure that same number of UAS constructs are used. In fig. 7, comparisons between overexpression and deletion don’t seem to have appropriate controls.  6. In overexpression analyses, the authors use a nuclear form of PTP61F. They should explain their rationale for this choice. They first use PTP61Fn2.1 and then later switch to PTP61Fn4.1 without explaining what these designations mean or why they are switching between them. 7. I found the depiction of GFP (-) area in Fig. 4E hard to follow. The genotype labels should indicate which genotypes are GFP+ versus GFP-. For example, in the second bar, the PTP61Fn2.1 expressing cells are GFP+, so the change in GFP- reflects the wildtype cell cluster size.  8. Following up on that, I think the details about double marked clones and autonomous versus non-autonomous effects might be lost on readers except for the view that specialize in this. A cartoon depicting genotypes in clones and arrows showing the hypotheses they are testing in a particular experiment might go far in making the data easier to follow (for example for lines 316-330). 9. Fig. 5 needs a wildtype control image (EAG>GFP) before panel A with labels highlighting what the structures are.  10. In section 2.6 the authors talk about their work looking at Jak-Stat signaling. They use both Stat92E-lacZ and Stat92E-GFP. Please explain why they switch between these. Was it simply to make the genetics easier or was there another reason? 11. Labels in Fig. 6D are far too small, impossible to read. Make sure genotype labels above blots match those used in legend. 12. The discussion is very long relative to the rest of the paper. If all the discussion is needed, perhaps break paragraphs apart a bit and include sub-headings? 13. In Fig. 8 the oval with PTP61F is too dark, lettering is hard to see and this is the 

Author Response

  1. Many of the figure legends (2, 3, 4) mistakenly say “wing imaginal discs” while eye discs are shown.

We apologise for this error – this has now been corrected.

  1. The authors initially examine clones in wing discs, then switch entirely to eyes. Was there a rationale for this? The two clones shown in the wing disc appear to be in the notum and hinge precursors. Were clones observed in the pouch domain?

The eye-antennal disc was chosen for analysis due to the availability of reliable tissue-specific Flipase reagents (ey-FLP). Since we observed a similar effect in the eye-antennal and wing discs using the hs-FLP (Figure 1), it is likely that the results we obtained in the eye-antennal disc are also relevant to other Drosophila imaginal epithelial tissues.

Yes, we also observed clones in the pouch domain.  Clones were observed in all regions of both the eye-antennal and wing discs, without exception. We have added clarification of this to the legend of Figure 1 (lines 164-166).

  1. 1E, F – the legend describes yellow bars. It would be clearer to say yellow rectangles. I presume the area of the rectangles reflect the size, rather than just location of the end of the rectangle?

Corrected in the figure legend.

The average size of the GFP--/GFP++ clone area ratio is only reflected by the end location of the rectangle, following the upper left axis of the graph. The inclusion of these rectangles is a stylistic choice, as the values represented are given in the text, and so they can be removed if the reviewer feels they detract from the figure.

  1. The authors in Fig. 1 show that loss of PTP61F gives clones a growth advantage over neighboring wildtype cells and use this fact as the starting point for their studies. This effect is not seen with the PTP61F RNAi in Fig. 2 & 3. The authors should point out that this effect is weaker and why they opted to use RNAi.

Yes, the Ptp61F-RNAi knockdown has a weaker phenotypic effect than the Ptp61F deletion.  We used RNAi lines as it is genetically very difficult to generate double scrib mutant PTP61F mutant clones as the genes are located on opposite chromosome arms of the 3rd chromosome.  

  1. Do the authors use Gal4 titration controls? In cases where numerous Gal4-driven transgenes are used and effects are compared, want to make sure that same number of UAS constructs are used. In fig. 7, comparisons between overexpression and deletion don’t seem to have appropriate controls.

Yes, GAL4 titration controls are used where possible. Although titration of GAL4 is theoretically an issue, practically we have not observed any significant effect on phenotypes when we use strong GAL4 drivers (eg eq-GAL4 and Tb-GAL4 in the MARCM system). In Figure 7, there are the same number of UAS elements for the Ptp61F deletion and the dlg-RNAi RasV12 control, whilst there is an additional UAS element in the UAS-Ptp61Fn expression relative to the dlg-RNAi RasV12 control. In our previous study (Willoughby et al., 2017 FEBS J, doi:10.1111/febs.14118, Figure 6), the size of the Eq domain due to the expression of UAS-PVRl from the eq driver, which caused tissue overgrowth, was not affected by another UAS transgene (UAS-Ptp61F-RNAi) although a robust increase in STAT expression was observed that was not seen with either transgene alone. This result shows that a second UAS element does not alter the overgrown Eq domain phenotype due of the expression of the UAS-PVRl transgene, and this is likely to also be the case with the dlg-RNAi RasV12 phenotype.  

In Figure 3, using the MARCM system (Tb-GAL4 driver) adding a second UAS element (UAS-Ptp61F-RNAi) did not affect the average clonal volume of UAS-STAT-RNAi expressing clones, showing that an additional UAS element is not an issue in reducing phenotype severity in this experiment.

In all the other figures and experiments we have used appropriate GAL4 titration controls.

  1. In overexpression analyses, the authors use a nuclear form of PTP61F. They should explain their rationale for this choice. They first use PTP61Fn2.1 and then later switch to PTP61Fn4.1 without explaining what these designations mean or why they are switching between them.

We used the nuclear form of PTP61F (isoform B) since it is a more potent variant of PTP61F as determined in our previous analysis.  Fn2.1 and Fn4.1 are transgenic UAS-PTP61Fn lines on the 2nd and 3rd chromosomes, respectively. We have added explanations regarding this to the text (line 303 and 449). This is also detailed mentioned in fly stock table (Supplementary Table 1).

  1. I found the depiction of GFP (-) area in Fig. 4E hard to follow. The genotype labels should indicate which genotypes are GFP+ versus GFP-. For example, in the second bar, the PTP61Fn2.1 expressing cells are GFP+, so the change in GFP- reflects the wildtype cell cluster size. 

We are interested in the effect of manipulating PTP61F expression in clones on the volume of the wild-type or scrib- (GFP-negative) tissue so the graph (Fig. 4E) plots the relative GFP-negative tissue volume. The white bar above the figures indicates that the Ptp61Fn-expressing tissue is GFP-positive.

  1. Following up on that, I think the details about double marked clones and autonomous versus non-autonomous effects might be lost on readers except for the view that specialize in this. A cartoon depicting genotypes in clones and arrows showing the hypotheses they are testing in a particular experiment might go far in making the data easier to follow (for example for lines 316-330).

We have added further explanation regarding this point to this section. We also believe the labelling included above each figure to be sufficient to explain the clonal genotypes, but we have also added text to the relevant figure legends highlighting these boxes and their colour-coding with regard to the clones being observed. Additionally, we have added cartoons to represent the GFP-status of the genotypes in each experiment and to highlight our hypotheses.

  1. 5 needs a wildtype control image (EAG>GFP) before panel A with labels highlighting what the structures are. 

We have added this figure to the image, circled the different structures, and explained our labelling in the figure legend.

  1. In section 2.6 the authors talk about their work looking at Jak-Stat signaling. They use both Stat92E-lacZ and Stat92E-GFP. Please explain why they switch between these. Was it simply to make the genetics easier or was there another reason?

The Stat92E-LacZ reporter was used since, in this experiment, GFP was present in the dlg-RNAi RasV12 stock (eyFLP ; UAS-Ras85DV12, UAS-dlg1RNAi (v41134) / + , tub-GAL80 ; Actin>CD2>GAL4, UAS-GFP).

  1. Labels in Fig. 6D are far too small, impossible to read. Make sure genotype labels above blots match those used in legend.

We have rearranged this figure to place the graphs below their corresponding western blot images, as well as numbered the tracks instead of the labelling to make it clearer to see.

  1. The discussion is very long relative to the rest of the paper. If all the discussion is needed, perhaps break paragraphs apart a bit and include sub-headings?

We believe that all of the discussion is necessary. We have added sub-headings to the Discussion to indicate the focus of various parts.

  1. In Fig. 8 the oval with PTP61F is too dark, lettering is hard to see and this is the 

We have lightened the colour of the oval backgrounding PTP61F.

Reviewer 2 Report

the manuscript is well written but need to address following issues before acceptance.

  1. introduction should be more specific and references are not cited properly. some places order of the referrences is missing in the manuscript.
  2. order of figure labelling is missing in text part.
  3. explain the figure in detail in figure legends and figure legends should be more elaborative
  4. correct figure 2 graph by reducing the bar label sizes and explain what is A, B, C in figure legends to all figures.
  5. Figure 4 is not visible completely.
  6. statistical error is missing in figure 6B&D
  7. correct the manuscript. a lot of grammatical errors.
  8. Authors has to show how the work is different from " PTP61F on RAS-STAT signaling in epithelial cells of drosophila" (https://doi.org/10.1111/febs.14118) and what novelty has been shown in the manuscript.

Author Response

  1. introduction should be more specific and references are not cited properly. some places order of the references is missing in the manuscript.

We disagree with this reviewer and feel that our introduction is specific to the topic and concise.  We have checked the references and made some corrections.

  1. order of figure labelling is missing in text part.

We do not understand the reviewer’s problem with the order of figure labelling in the text.

  1. explain the figure in detail in figure legends and figure legends should be more elaborative

We have tried to be provide sufficient detail but be concise in the figure legends. We do not understand the reviewer’s specific problem with the figure legend details, but have made some edits to explain the figures more thoroughly.

  1. correct figure 2 graph by reducing the bar label sizes and explain what is A, B, C in figure legends to all figures.

We have made edits to describe in more detail the figure panels in each figure.

  1. Figure 4 is not visible completely.

We have realigned Figure 4 so that it is now inside the page field.

  1. statistical error is missing in figure 6B&D

The experiment represents one biological replicate and therefore statistical errors cannot be added to the figure.

  1. correct the manuscript. a lot of grammatical errors.

We have proofread our manuscript for typographical and grammatical errors and cannot find any. Please provide more details on where the grammatical errors are in our manuscript.

  1. Authors has to show how the work is different from " PTP61F on RAS-STAT signaling in epithelial cells of drosophila" (https://doi.org/10.1111/febs.14118) and what novelty has been shown in the manuscript.

Our current manuscript is focussed on determining the role of PTP61F in cell competition and cooperative tumourigenesis, not just on which signalling pathways it controls. The novelty of our manuscript is that it reveals that PTP61F plays a role in cell competition, and in cell-polarity impaired cell competition it plays a more potent role non-cell autonomously than cell autonomously. Moreover, we show that PTP61F acts as a tumour suppressor in cell-polarity-impaired Ras-driven tumourigenesis. 

Round 2

Reviewer 2 Report

the authors corrected the manuscript into acceptable format.